# Adaptive Thermal Imaging Signal Analysis for Real-Time Non-Invasive Respiratory Rate Monitoring

**DOI:** 10.3390/s26010278

**Published:** 2026-01-01

**Authors:** Riska Analia, Anne Forster, Sheng-Quan Xie, Zhiqiang Zhang

**Affiliations:** 1School of Electronic and Electrical Engineering, University of Leeds, Leeds LS2 9LN, UK; elran@leeds.ac.uk (R.A.); s.q.xie@leeds.ac.uk (S.-Q.X.); 2Department of Electrical Engineering, Politeknik Negeri Batam, Batam 29461, Indonesia; 3Academic Unit for Ageing and Stroke Research, Leeds Institute of Health Sciences, University of Leeds, Leeds LS2 9JT, UK; a.forster@leeds.ac.uk

**Keywords:** adaptive, contactless, embedded edge hardware, inter-breath intervals, non-invasive sensing, respiratory rate monitoring, real-time, thermal imaging

## Abstract

(1) Background: This study presents an adaptive, contactless, and privacy-preserving respiratory-rate monitoring system based on thermal imaging, designed for real-time operation on embedded edge hardware. The system continuously processes temperature data from a compact thermal camera without external computation, enabling practical deployment for home or clinical vital-sign monitoring. (2) Methods: Thermal frames are captured using a 256×192 TOPDON TC001 camera and processed entirely on an NVIDIA Jetson Orin Nano. A YOLO-based detector localizes the nostril region in every even frame (stride = 2) to reduce the computation load, while a Kalman filter predicts the ROI position on skipped frames to maintain spatial continuity and suppress motion jitter. From the stabilized ROI, a temperature-based breathing signal is extracted and analyzed through an adaptive median–MAD hysteresis algorithm that dynamically adjusts to signal amplitude and noise variations for breathing phase detection. Respiratory rate (RR) is computed from inter-breath intervals (IBI) validated within physiological constraints. (3) Results: Ten healthy subjects participated in six experimental conditions including resting, paced breathing, speech, off-axis yaw, posture (supine), and distance variations up to 2.0 m. Across these conditions, the system attained a MAE of 0.57±0.36 BPM and an RMSE of 0.64±0.42 BPM, demonstrating stable accuracy under motion and thermal drift. Compared with peak-based and FFT spectral baselines, the proposed method reduced errors by a large margin across all conditions. (4) Conclusions: The findings confirm that accurate and robust respiratory-rate estimation can be achieved using a low-resolution thermal sensor running entirely on an embedded edge device. The combination of YOLO-based nostril detector, Kalman ROI prediction, and adaptive MAD–hysteresis phase that self-adjusts to signal variability provides a compact, efficient, and privacy-preserving solution for non-invasive vital-sign monitoring in real-world environments.

## 1. Introduction

Continuous monitoring of respiratory rate (RR) has become one of the most important vital signs in both clinical and home care settings. RR plays a crucial role in assessing a physiological condition of the patient, especially during clinical deterioration [1,2,3]. Early detection through continuous RR tracking enables timely intervention, particularly for high-risk populations [4,5,6,7]. In long-lie conditions following a fall, continuous respiratory monitoring offers valuable physiological information for early detection and timely assistance [8,9,10].

Several conventional approaches have been employed in respiratory monitoring studies. These typically involve physical contact with the patient, including chest bands, nasal cannulas, or spirometry devices. While these tools are clinically validated, they often cause discomfort and may interfere with natural breathing behavior, especially during sleep or prolonged observation periods [11,12,13,14]. A non-contact alternative method, on the other hand, has shown promising results in estimating RR, which could be one of the solutions to overcome these challenges. A recent study developed a non-contact system using radar-based techniques, acoustic sensors, and camera-based methods such as RGB or thermal imaging [15,16,17,18].

Among these non-contact modalities, thermal imaging presents distinct advantages for RR estimation. It enables the capture of temperature variations generated by inhaled and exhaled air without physical contact. These thermal fluctuations, observed around the nostril or mouth region, offer a natural and unobtrusive signal source for respiratory analysis [19,20]. This makes thermal-based systems particularly well-suited for continuous monitoring in privacy-sensitive environments such as bedrooms or elder care facilities.

Despite its promise, thermal-based respiratory monitoring still faces several technical challenges. Accurate detection and tracking of the nostril region is often hindered by the low spatial resolution of thermal cameras, which complicates region of interest (ROI) localization [21,22]. Furthermore, thermal signals are susceptible to noise introduced by subject movement, head rotations, and ambient temperature changes, all of which can degrade signal quality and affect RR estimation accuracy [22,23]. Moreover, many existing implementations rely on frequency-domain analysis or computationally expensive deep learning models, which limits their real-time feasibility on embedded platforms [19,24,25,26,27].

Recent biomedical monitoring systems have increasingly shifted toward embedded edge computing due to its advantages in latency, privacy, and deployment feasibility. Unlike cloud-based processing, which introduces transmission delays and raises concerns over sensitive health data exposure, edge computation allows all inference to occur locally on the device. This enables real-time responsiveness and preserves subject privacy, which are essential for continuous respiratory monitoring [28,29,30,31,32]. However, most existing thermal and camera-based respiratory monitoring studies still rely on offline processing pipelines due to their high computational requirements, limiting their applicability for real-time embedded deployment.

Therefore, current thermal-based RR approaches still leave three critical gaps unaddressed: (i) the lack of a robust nostril-specific localization strategy for low-resolution thermal imagery, resulting in unstable ROI tracking; (ii) the absence of computational optimization needed for real-time deployment on embedded edge devices; and (iii) limited robustness of time-domain phase detection, which remains sensitive to motion-induced disturbances, amplitude variability, and thermal drift. These unmet needs motivate the development of a thermal-specific, computation-efficient, and motion-resilient respiratory monitoring framework suitable for continuous operation in real-world environments.

To overcome these limitations, this study introduces a fully automated, privacy-preserving thermal-imaging system for real-time respiratory-rate monitoring on embedded edge hardware. The system begins with a thermal YOLO-based model to locate the nostril region as a small-object bounding box; this box defines the region of interest (ROI) from which an airflow-related temperature signal is extracted using the coldest pixel within the ROI, reflecting inhalation–exhalation temperature modulation. To reduce computational cost at the detector stage, an adaptive frame skipping (stride = 2) with Kalman prediction is applied so that the YOLO detector runs at half the nominal frequency while a Kalman tracker tracks the nostril bounding-box (bbox) between detections, preserving continuity and suppressing motion artifacts. The YOLO-based nostril detector was implemented using the YOLOv8n model, which naturally supports small-object detection and operates efficiently on low-resolution thermal imagery.

Respiratory-rate estimation begins with breathing-phase detection on the stabilized ROI signal using an adaptive hysteresis state machine driven by velocity-based thresholds. These thresholds are derived from the median absolute deviation (MAD) and integrated with a flicker-suppression mechanism to maintain signal stability during head movement and other motion disturbances. The resulting stable breathing-phase sequence is then used to determine inter-breath intervals (IBI), from which the respiratory rate (RR) is calculated.

The main contributions of this work are: (i) a thermal-specific YOLO-based nostril detector designed for small-object detection in low-resolution 256 × 192 thermal imagery, overcoming ROI instability common in prior thermal RR studies, (ii) a detector-centric frame-skipping mechanism (stride = 2) integrated with Kalman ROI prediction, reducing detection computation by 50% while maintaining spatial continuity and enabling real-time embedded operation, (iii) an adaptive time-domain respiratory-phase detection approach that combines median–MAD thresholds, hysteresis, and flicker suppression to achieve robust segmentation under motion, drift, and amplitude variability, without relying on frequency-domain analysis, (iv) a fully on-device respiratory-rate monitoring pipeline, running entirely on an NVIDIA Jetson Orin Nano without cloud processing, ensuring privacy preservation and practical deployment feasibility for long-term ambient monitoring, and (v) a comprehensive evaluation across six real-world conditions (resting, paced breathing, soft speech, off-axis yaw, distance variation up to 2.0 m, and supine posture), demonstrating clinically acceptable accuracy (overall MAE 0.57 ± 0.36 BPM), outperforming previously reported thermal-based contactless RR systems. To guide the development of this work, the following research questions are formulated:

RQ1: Can a low-resolution thermal camera, combined with automated nostril tracking, provide accurate and reliable respiratory-rate estimation across diverse real-world conditions?

RQ2: How can adaptive signal-processing strategies, such as MAD-based breathing-phase detection and IBI validation, improve robustness against facial movement, off-axis orientation, and varying thermal contrast?

RQ3: Is the proposed approach computationally lightweight enough to operate in real time on an embedded edge device without compromising accuracy?

These questions motivate the system design and experimental evaluation presented in the remainder of this paper.

The remainder of this paper is structured as follows: related work is reviewed in Section 2; Section 3 describes the proposed methodology, including data acquisition and model architecture; Section 4 presents experimental results; Section 5 discusses system performance and deployment feasibility; and Section 6 concludes the paper with a summary and directions for future work.

## 2. Related Work

Among the human vital signs, respiratory rate (RR) is widely recognized as a critical indicator of physiological stability. In estimating the RR, thermal imaging has emerged as a promising non-contact and privacy-preserving modality. Unlike contact-based methods that require direct attachment to the body, thermal cameras detect temperature variations caused by airflow during inhalation and exhalation, typically around the nostrils or mouth. These thermal fluctuations form a natural and unobtrusive signal source for respiratory analysis, particularly suitable for continuous monitoring in both clinical and home care settings [33,34,35].

A variety of methods have been proposed to extract respiratory signals from thermal video data. Earlier approaches relied on manual region-of-interest (ROI) selection and simple pixel averaging, which were limited robustness under motion or occlusion. More recent developments introduced computer vision and deep learning techniques for automated ROI localization, including three-dimensional convolutional neural networks, detection transformers, and single-shot detectors such as YOLO and SSD [26,27,36]. Facial landmark–based approaches have also been used to improve stability under head motion or partial occlusion [37,38]. Other studies align RGB landmarks with thermal images [22] or extract thermal–motion data to detect breathing regions even when facial features are obscured by masks or bedding [23,39,40]. However, small-object detection of the nostril region in low-resolution thermal frames remains challenging.

Once the ROI is detected respiratory signals are typically obtained by tracking temperature variations over time. To enhance signal quality, various filtering methods such as Butterworth, Hampel, and Savitzky–Golay have been employed, along with adaptive decomposition techniques like the Hilbert–Huang Transform [36,41,42]. Other preprocessing strategies, including histogram equalization, optimal quantitation, and super-resolution, have been applied to compensate for the low resolution of compact thermal cameras [43]. Moreover, some works directly apply deep models to thermal sequences, learning temporal breathing patterns without explicit ROI tracking [19,26]. Many studies estimate RR using dominant spectral components via Fourier, synchro-squeezed, or autocorrelation analysis [27,34,42], but frequency-domain approaches can be sensitive to noise, motion artifacts, and ambient temperature drift. Few works incorporate explicit flicker-suppression or median-absolute-deviation-based adaptive thresholds in the breathing phase logic to stabilize the signal during head movement.

Beyond algorithmic advances, several recent works emphasize that embedded edge computing has become a key requirement for biomedical sensing systems [28,29]. On-device processing reduces communication overhead, enhances data security, and enables deployment in resource-constrained settings where continuous cloud connectivity is impractical [30,31,32]. Despite the increasing adoption of edge-based architectures, existing thermal RR methods seldom address the computational constraints of embedded hardware, with many relying on high-resolution cameras or offline deep models. This gap further motivates the development of a lightweight and fully embedded respiratory-rate monitoring pipeline.

Based on the gaps outlined above, this paper proposes an adaptive, real-time thermal respiratory monitoring system for embedded deployment. The contributions comprise a thermal-specific YOLO-based detector for nostril localization, a detector-stage frame-skipping scheme with Kalman prediction to halve detection frequency while preserving ROI continuity, and an adaptive MAD–hysteresis phase detection framework with flicker suppression for motion-robust, physiologically consistent respiratory-rate estimation, all validated on-device under privacy-preserving constraints.

## 3. Materials and Methods

### 3.1. System Overview

The proposed system converts raw thermal video frames into RR estimation through a sequence of tightly coupled modules, as illustrated in Figure 1. Thermal frames are first captured by a low-resolution thermal camera that provides two synchronous outputs: an thermal imagery frame and a thermal data. A YOLO-based detector localizes the nostril on the thermal imagery every second frame, and the resulting ROI is projected to the thermal mapping image; between detections, a Kalman prediction updates the ROI directly in thermal coordinates. The thermal data are decoded into per-pixel temperatures to form a calibrated temperature map; the tracked bounding box crops this map, and the coldest pixel temperature per frame serves as a one-dimensional airflow-related signal. This signal is band-pass filtered in the 0.08 to 0.7 Hz range with a fourth-order zero-phase Butterworth filter and analyzed in the time domain using velocity estimates with median-absolute-deviation (MAD)–derived thresholds. An adaptive hysteresis state machine with a minimum dwell of 0.15 s produces inhale, exhale, and hold phases. Phase transitions yield inter-breath intervals (IBI) that are validated within a physiologic range and converted to breaths per minute (BPM), then stabilized by short weighted averaging and exponential moving averaging.

### 3.2. Thermal Camera Acquisition

The initial captured frame from a thermal camera consists of both image and thermal information; therefore, splitting image and thermal information is necessary as the first step in processing thermal camera acquisition data. The process of thermal acquisition is illustrated in Figure 2. The initial capture frame can be expressed as(1)$$I_{\mathrm{raw}}\in Z_{256}^{H\times 2W},$$
where *H* and *W* represent the height and width of a single modality, and$$Z_{256}=\left\{0,1,\ldots ,255\right\}$$
denotes the set of 8-bit integer values corresponding to raw pixel intensities. The raw frame is divided into two separate streams:(2)$$I_{\mathrm{raw}}\overset{\mathrm{split}}{\to }\left\{I_{\mathrm{img}}\in Z_{256}^{H\times W},D_{\mathrm{th}}\in Z_{256}^{H\times W\times 2}\right\},$$
with $I_{\mathrm{img}}$ denoting the YUV image data and $D_{\mathrm{th}}$ denoting the paired bytes of thermal information. Each stream is subsequently processed within the same acquisition cycle but along independent pathways as presented in Figure 2: the $I_{\mathrm{img}}$ branch is converted into a color-mapped thermal image for visualization, whereas the $D_{\mathrm{th}}$ branch is decoded into pixel-wise temperature values, creating the quantitative temperature map corresponding to each thermal frame.

#### 3.2.1. Image Data Processing

The image stream $I_{\mathrm{img}}$ undergoes a sequence of preprocessing operations to produce a heatmap suitable for visualization. The raw frames, initially captured in YUV format, are first converted into an RGB representation,(3)$$I_{\mathrm{RGB}}=\mathrm{YUV}2\mathrm{RGB}\left(I_{\mathrm{img}}\right),$$
the $I_{\mathrm{RGB}}$ denotes the color image obtained from $I_{\mathrm{img}}$. To enhance visual clarity, a linear contrast adjustment is then applied, expressed as(4)$$I_{\mathrm{RGB}}^{\prime }=\alpha \times I_{\mathrm{RGB}},\alpha =1.0,$$
the α denote as the contrast scaling factor. In this work, α is fixed to unity, implying no additional scaling beyond the raw dynamic range.

Subsequently, the frame is spatially upscaled by bicubic interpolation:(5)$$I_{\mathrm{RGB}}^{\prime \prime }=\mathrm{Resize}\left(I_{\mathrm{RGB}}^{\prime },W^{\prime },H^{\prime }\right),W^{\prime }=3W,H^{\prime }=3H,$$
where Resize(·) denotes the interpolation operator, and $\left(W^{\prime },H^{\prime }\right)$ are the target spatial dimensions set to three times the original resolution (W,H). Finally, the enhanced frame is mapped into a false-color domain for visualization through(6)$$I_{\mathrm{heatmap}}=\mathrm{ColorMap}\left(I_{\mathrm{RGB}}^{\prime \prime }\right),$$
the ColorMap(·) maps the intensity distribution of $I_{\mathrm{RGB}}^{\prime \prime }$ into a perceptually enhanced heatmap representation for display visualization.

#### 3.2.2. Thermal Data Processing

In the thermal data processing stage, pixel-wise temperatures are decoded from the paired bytes. Each thermal frame stores temperature values in two consecutive 8-bit values corresponding to the most significant byte (MSB) and least significant byte (LSB), which are combined and linearly calibrated to form the quantitative temperature map shown in Figure 3. For each pixel coordinate (x,y) with(7)x∈{0,…,W−1}, y∈{0,…,H−1},
let H(x,y) and L(x,y) be the MSB and LSB, respectively, i.e., $H,L\in Z_{256}$. The raw temperature proxy is reconstructed as(8)$$R\left(x,y\right)=\frac{256H(x,y)+L(x,y)}{64}-273.15.$$

The combined value (256H(x,y)+L(x,y)) represents raw temperature data encoded in Kelvin ×64 according to the manufacturer’s format. Dividing by 64 converts this to Kelvin, and subtracting 273.15 yields the temperature in degrees Celsius, resulting in the calibrated temperature map R(x,y). A linear calibration model is subsequently applied to compensate for sensor bias:(9)T(x,y)=αR(x,y)+β,
where α is the calibration gain (scaling factor) and β is the calibration offset (bias in °C). The calibrated value T(x,y) corresponds to the corrected temperature at pixel (x,y). And the maximum temperature within a frame is then localized as(10)$$\left(x^{*},y^{*}\right)=\underset{\left(x,y\right)}{arg max}T\left(x,y\right),T_{\max}=T\left(x^{*},y^{*}\right),$$
where $\left(x^{*},y^{*}\right)$ indicates the pixel coordinates of the hottest point in the frame and $T_{\max}$ is its corresponding temperature. For visualization, the calibrated temperature field is interpolated to generate a thermal map:(11)$$T_{\mathrm{map}}=\mathrm{Resize}\left(T\left(x,y\right),W^{\prime },H^{\prime }\right),$$
where $\mathrm{Resize}(\cdot ,W^{\prime },H^{\prime })$ denotes a spatial interpolation operator mapping the original temperature matrix of size H×W to a new resolution $H^{\prime }\times W^{\prime }$ for display purposes. The resulting $T_{\mathrm{map}}$ provides the visualization branch of the processing pipeline.

### 3.3. ROI Localization and Temperature Feature Extraction

#### 3.3.1. YOLO-Based Nostril Detection

Respiratory monitoring based on wearable sensors is often limited by discomfort, movement artifacts, and the need for frequent recalibration [44,45]. RGB video methods offer a non-contact alternative but remain highly sensitive to illumination changes, motion artifacts, and computational overhead [46,47,48]. Thermal imaging provides a more suitable modality because it is independent of lighting conditions and relatively robust to minor head movements, making it advantageous for continuous monitoring. Yet the low resolution and limited texture of thermal frames make nostril localization challenging. To overcome this difficulty, a YOLO-based detection model was adopted, as illustrated in Figure 4. YOLO was chosen for its balance between detection accuracy and computational efficiency, which makes it suitable for real-time deployment on embedded edge hardware. Unlike RGB data, thermal frames contain only coarse temperature gradients and lack color cues, which reduces the effectiveness of standard feature extraction; accordingly, architectural and training adaptations were required.

In this work, YOLOv8n was selected as the object detection framework for its computational efficiency and suitability to real-time thermal imaging. The lightweight model architecture enables effective feature extraction while maintaining low computational cost, which is appropriate for the relatively simple thermal domain where nostril regions are primarily defined by local hot-cold gradients rather than complex textures. The detection head maintains high-resolution feature maps, improving the sensitivity of the model to small ROI that occupy less than five percent of the thermal frame. In parallel, the training procedure was tailored to the thermal modality. A dataset of 7958 annotated thermal images (7113 training, 563 validation, 282 testing) was assembled, incorporating variations in head orientation, distance, and partial occlusion. Data augmentation strategies avoided color-based transformations and instead emphasized brightness and contrast adjustments, Gaussian noise injection, and mild geometric perturbations to reproduce sensor variability and natural subject motion.

For deployment, the trained detector is executed using the Ultralytics YOLO runtime on the embedded device, without reliance on external deep-learning frameworks or GPU acceleration. To reduce computational load, detection is performed on every even frame, and a Kalman filter predicts the ROI on intermediate frames. With frame index $k\in Z_{\geq 0}$ and camera rate $f_{\mathrm{cam}}$ (Hz), the detection schedule is(12)$$S_{\det}=\left\{k|k\mathrm{mod}2=0\right\}\left(\mathrm{stride}s=2\right),$$
which yields the effective detection rate(13)$$f_{\det}=\frac{f_{\mathrm{cam}}}{s}=\frac{f_{\mathrm{cam}}}{2}.$$

The ROI used at frame *k* is obtained from the detector when $k\in S_{\det}$ (and a detection exists) and from the Kalman prediction otherwise:(14)$${\mathrm{ROI}}_{k}=\left\{\begin{array}{cc} {\mathrm{ROI}}_{k}^{\det}, & k\in S_{\det}\mathrm{and}a\mathrm{detection}\mathrm{exists},\\ {\mathrm{ROI}}_{k}^{\mathrm{trk}}, & \mathrm{otherwise}. \end{array}\right.$$

#### 3.3.2. Kalman Filter Tracking

Since YOLO-based generates nostril detections only on even frames, the Kalman filter predicts the ROI on odd frames and whenever an even–frame detection is unavailable. To provide stable localization of the nostril region, an eight–dimensional Kalman filter jointly estimates the bounding–box position and its temporal dynamics. The state vector is defined as(15)$$x_{k}={\left[c_{x},c_{y},w,h,{\dot{c}}_{x},{\dot{c}}_{y},\dot{w},\dot{h}\right]}^{⊤},$$
where $\left(c_{x},c_{y}\right)$ are the bounding–box center coordinates (pixels), (w,h) its width and height (pixels), and the dotted variables the corresponding temporal velocities.

The temporal evolution of the state follows a discrete constant–velocity model:(16)$$x_{k}=Fx_{k-1}+w_{k-1},\;w_{k-1}∼N\left(0,Q\right),$$
where F is the 8×8 transition matrix, $w_{k-1}$ the process noise, and Q its covariance. Using the frame interval $\Delta t=1/f_{\mathrm{cam}}$, the transition matrix is(17)$$F=\left[\begin{array}{cc} I_{4} & \Delta tI_{4}\\ 0_{4} & I_{4} \end{array}\right],$$
with $I_{4}$ the 4×4 identity matrix and $0_{4}$ the 4×4 zero matrix. The choice of the constant-velocity state–space model in Equations (Equation 15) and (Equation 16) follows standard formulations widely used in visual object tracking, as it provides a minimal yet sufficiently expressive representation of bounding-box motion [49,50]. The transition matrix F is block-structured with identity and $\Delta tI_{4}$ submatrices, which yields eigenvalues equal to 1. Consequently, the discrete-time system is marginally stable, as expected for constant-velocity motion models; the continuous-time Hurwitz condition does not directly apply in this setting. Since the Kalman filter is employed solely for state estimation rather than control, controllability is not required. The pair (F,H) is observable, as the associated observability matrix has full rank for any Δt>0, ensuring that all components of the state vector, including position, size, and their velocities are inferable from the detector measurements.

At each frame, the YOLO–based detector produces a bounding box with corners $\left(x_{1},y_{1}\right)$ (top–left) and $\left(x_{2},y_{2}\right)$ (bottom–right). This is converted into the measurement vector(18)$$z_{k}={\left[\frac{x_{1}+x_{2}}{2},\frac{y_{1}+y_{2}}{2},x_{2}-x_{1},y_{2}-y_{1}\right]}^{⊤},$$
which contains the observed center position and box dimensions. The measurement model is(19)$$z_{k}=Hx_{k}+v_{k},\;v_{k}∼N\left(0,R_{k}\right),\;H=\left[I_{4}0_{4\times 4}\right].$$

Let $d_{k}\in \left\{0,1\right\}$ indicate whether a detector output exists at frame *k* (on scheduled even frames). The measurement–use indicator is(20)$$m_{k}=\left\{\begin{array}{cc} 1, & \mathrm{if}\;k\;\mathrm{mod}\;2=0\;\mathrm{and}\;d_{k}=1,\\ 0, & \mathrm{otherwise}, \end{array}\right.$$
and a fixed measurement covariance is used(21)$$R_{k}=R.$$

The recursion runs on every frame. Prediction:(22)$${\hat{x}}_{k|k-1}=F{\hat{x}}_{k-1},\;P_{k|k-1}=FP_{k-1}F^{⊤}+Q.$$

Update (only if $m_{k}=1$):(23)$$y_{k}=z_{k}-H{\hat{x}}_{k|k-1},\;S_{k}=HP_{k|k-1}H^{⊤}+R_{k},$$(24)$$K_{k}=P_{k|k-1}H^{⊤}S_{k}^{-1},\;{\hat{x}}_{k}={\hat{x}}_{k|k-1}+K_{k}y_{k},$$(25)$$P_{k}=\left(I-K_{k}H\right)P_{k|k-1}.$$

If $m_{k}=0$, the prediction becomes the current estimate. The output box is reconstructed as(26)$$x_{1}=c_{x}-\frac{w}{2},\;y_{1}=c_{y}-\frac{h}{2},\;x_{2}=c_{x}+\frac{w}{2},\;y_{2}=c_{y}+\frac{h}{2}.$$
This even–odd schedule halves detector invocations, provides ROI estimates for skipped frames via Kalman prediction, and preserves temporal continuity under short dropouts, head motion, or partial occlusion.

#### 3.3.3. Temperature Extraction

Once the nostril ROI is localized by the detection–tracking pipeline, the calibrated thermal image at frame *k* is treated as a discrete grid $T_{k}\left[x,y\right]$ (temperature in °C at pixel (x,y)). The ROI is the integer-indexed set(27)$${\mathrm{ROI}}_{k}=\left\{\left(x,y\right)\in Z^{2}|x_{1}^{\left(k\right)}\leq x\leq x_{2}^{\left(k\right)},\;y_{1}^{\left(k\right)}\leq y\leq y_{2}^{\left(k\right)}\right\},$$
where $\left(x_{1}^{\left(k\right)},y_{1}^{\left(k\right)}\right)$ and $\left(x_{2}^{\left(k\right)},y_{2}^{\left(k\right)}\right)$ are the top-left and bottom-right corners of the bounding box at frame *k*. The representative nostril temperature for frame *k* is the minimum within the ROI,(28)$${\hat{T}}_{k}=\min_{\left(x,y\right)\in {\mathrm{ROI}}_{k}}T_{k}\left[x,y\right],$$
and the pixel attaining this minimum is recorded as(29)$$p_{k}^{*}=\underset{\left(x,y\right)\in {\mathrm{ROI}}_{k}}{\arg\min}T_{k}\left[x,y\right].$$

Selecting the coldest pixel yields a physiologically consistent proxy of airflow, as inhalation introduces cooler ambient air whereas exhalation releases warmer expired air. Across frames, the extracted temperatures form the scalar sequence(30)$$S={\left\{{\hat{T}}_{k}\right\}}_{k=0}^{N-1},$$
where *N* is the number of samples retained in the observation buffer. A sliding buffer of approximately 20 s (i.e., $N\approx \lceil 20f_{\mathrm{cam}}\rceil$) preserves multiple respiratory cycles while maintaining responsiveness for real-time monitoring. This raw nostril-temperature signal is then used for band-pass filtering, phase detection, and respiratory-rate estimation. Figure 5 illustrates the extraction result: panel (a) shows the cropped thermal ROI with the coldest-temperature pixel at each frame, and panel (b) shows the resulting raw sequence S, whose cyclic oscillations align with inhalation (cooling) and exhalation (warming) events.

### 3.4. Adaptive Breathing Phase Detection and Respiratory Rate Calculation

#### 3.4.1. Adaptive Breathing Phase Detection

The breathing phase pipeline starts from the illustration in Figure 6: inhalation cools the nostril surface, whereas exhalation warms it. After ROI extraction, each frame *k* provides a scalar sample $\hat{T}\left[k\right]$ (in °C). With camera frame rate $f_{\mathrm{cam}}$ (Hz), samples occur at $t_{k}=k/f_{\mathrm{cam}}$; equivalently,(31)$$\hat{T}\left[k\right]=\hat{T}\left(t_{k}\right)+\varepsilon_{k},$$
where $\varepsilon_{k}$ denotes measurement noise. The processing that follows operates on the discrete sequence $\left\{\hat{T}\left[k\right]\right\}$. The raw sequence shows alternating cooling and warming, but is also affected by drift and noise. To describe its expected structure and guide filter design, it is convenient to write the quasi-periodic model(32)$$\hat{T}\left(t\right)=T_{0}+A\sin\left(2\pi f_{\mathrm{RR}}t+\phi \right)+d_{\mathrm{low}}\left(t\right)+n_{\mathrm{high}}\left(t\right),$$
where $T_{0}$ is the baseline temperature, *A* the oscillation amplitude, $f_{\mathrm{RR}}$ the respiratory frequency (Hz), ϕ the phase, $d_{\mathrm{low}}\left(t\right)$ a slow drift term, and $n_{\mathrm{high}}\left(t\right)$ high-frequency noise. This model is illustrative; the digital operations below use the sampled signal $\hat{T}\left[k\right]$.

Since respiration lies in the 0.08–0.7 Hz band, a 4th-order Butterworth band-pass filter with cutoffs at 0.08 and 0.7 Hz is applied. The filter is implemented with forward–backward recursion to ensure a zero-phase response, thereby preserving the temporal integrity of breathing cycles. The band-pass output satisfies the standard IIR difference equation:(33)$${\hat{T}}_{\mathrm{bp}}\left[n\right]=\sum_{m=0}^{M}b_{m}\hat{T}\left[n-m\right]-\sum_{j=1}^{N}a_{j}{\hat{T}}_{\mathrm{bp}}\left[n-j\right],$$
where $b_{m}$ and $a_{j}$ are the Butterworth coefficients. This passband covers 5–42 BPM and suppresses baseline drift and high-frequency noise, preserving breath timing. Building on the zero-phase band-passed output ${\hat{T}}_{\mathrm{bp}}\left[n\right]$, the local heating/cooling trend is quantified using a rectangular moving-average window of length *W*. The discrete velocity surrogate is defined as the difference between two adjacent moving averages:(34)$$v\left[n\right]=\left(r_{W}*{\hat{T}}_{\mathrm{bp}}\right)\left[n\right]-\left(r_{W}*{\hat{T}}_{\mathrm{bp}}\right)\left[n-W\right],$$
where “∗” denotes discrete convolution and $r_{W}\left[k\right]$ is the rectangular (uniform) kernel(35)$$r_{W}\left[k\right]=\left\{\begin{array}{cc} \frac{1}{W}, & 0\leq k\leq W-1,\\ 0, & \mathrm{otherwise}. \end{array}\right.$$

An equivalent summation form is(36)$$v\left[n\right]=\frac{1}{W}\sum_{k=0}^{W-1}{\hat{T}}_{\mathrm{bp}}\left[n-k\right]-\frac{1}{W}\sum_{k=W}^{2W-1}{\hat{T}}_{\mathrm{bp}}\left[n-k\right],$$
under the zero-phase design, the sign of v[n] aligns with physiology:(37)v[n]>0⇒exhalation(warming), v[n]<0⇒inhalation(cooling),
in implementation, a small window (e.g., W=3 samples) attenuates frame-to-frame jitter while preserving phase timing within the 5–42 BPM operating range.

To normalize across subjects and amplitudes, an adaptive, data-dependent threshold is derived from the most recent *L* velocity samples. Let the length-*L* window be(38)$$V_{n}≜\left\{v\left[n-i\right]|i=0,1,\ldots ,L-1\right\}.$$
The location statistic and dispersion are defined by the sample median and the median absolute deviation (MAD):(39)$$m_{v}\left[n\right]=\mathrm{median}\left(V_{n}\right),$$(40)$${\mathrm{MAD}}_{v}\left[n\right]={\mathrm{median}}_{i\in \{0,\ldots ,L-1\}}\left|v\left[n-i\right]-m_{v}\left[n\right]\right|.$$
A symmetric, scale-invariant threshold is then(41)$$\theta \left[n\right]=\alpha {\mathrm{MAD}}_{v}\left[n\right]+\varepsilon ,$$
with sensitivity coefficient α>0 (e.g., α=0.6), a short history length *L* (e.g., 15–25 samples), and a small ε>0 to avoid degeneracy when variability is minimal. The threshold is applied symmetrically around zero to map velocity to the physiological phase:(42)v[n]≥θ[n]⇒exhalation(warming), v[n]≤−θ[n]⇒inhalation(cooling).Optionally, a minimum dwell converts threshold crossings into stable segments; with sampling frequency $f_{\mathrm{cam}}$, the dwell length in samples is(43)$$N_{\min}=\left\lceil t_{\min}f_{\mathrm{cam}}\right\rceil ,$$
ensuring physiologically plausible durations within the 5–42 BPM operating range.

To convert the thresholded velocity into stable phase labels, a hysteretic state machine with a minimum dwell is employed. Let the instantaneous phase label be(44)s[n]∈{−1,0,+1},
representing Inhalation (−1), Neutral/Hold (0), and Exhalation (+1). The dwell requirement expressed in samples Equation (Equation 43), ensuring physiologically plausible segment durations within the 5–42 BPM operating range. Transitions are gated by the elapsed persistence of the current state. Let τ[n−1] denote the number of consecutive samples that state s[n−1] has persisted up to time n−1. The phase update is(45)$$s\left[n\right]=\left\{\begin{array}{cc} +1, & \mathrm{if}v\left[n\right]\geq \theta \left[n\right]\wedge \tau \left[n-1\right]\geq N_{\min},\\ -1, & \mathrm{if}v\left[n\right]\leq -\theta \left[n\right]\wedge \tau \left[n-1\right]\geq N_{\min},\\ s[n-1], & \mathrm{otherwise}. \end{array}\right.$$

The persistence counter is updated recursively as(46)$$\tau \left[n\right]=\left\{\begin{array}{cc} 0, & \mathrm{if}s[n]\neq s[n-1],\\ \tau [n-1]+1, & \mathrm{if}s[n]=s[n-1]. \end{array}\right.$$
Optionally, brief near-threshold fluctuations may be represented as a neutral state to emphasize ambiguity around zero velocity:(47)(optional) s[n]←0 if |v[n]|<θ[n].
This hysteretic formulation suppresses chatter from transient perturbations while preserving accurate timing of inhalation and exhalation transitions.

Following the hysteretic phase labeling, brief flicker patterns of the form A→B→A are suppressed by merging the short middle segment into its flanking phase. Here A,B∈{−1,0,+1} denote the phase labels in Equation (Equation 44). Let ${\left\{n_{i}\right\}}_{i\geq 0}$ denote the ordered change-points of s[n],(48)$$n_{0}=0,\;n_{i+1}=\min\left\{n>n_{i}|s\left[n\right]\neq s\left[n-1\right]\right\},$$
and define the *i*-th segment state and duration by(49)$$q_{i}≜s\left[n\right]\;\mathrm{for}\;\mathrm{all}\;n\in \left[n_{i},n_{i+1}-1\right],\;L_{i}≜n_{i+1}-n_{i}.$$
With sampling frequency $f_{\mathrm{cam}}$, the sample-based consolidation threshold is(50)$$N_{c}=\left\lceil \tau_{c}f_{\mathrm{cam}}\right\rceil ,$$
where $\tau_{c}$ is a short duration (e.g., $\tau_{c}=0.3s$) chosen to reject physiologically implausible micro-segments within 5–42 BPM.

The consolidation rule replaces the short intermediate phase *B* (i.e., $q_{i}$) by its flanking phase *A* (i.e., $q_{i-1}=q_{i+1}$) whenever an A→B→A pattern occurs and the intermediate duration is below $N_{c}$:(51)$$\mathrm{if}\;q_{i-1}=q_{i+1}\wedge q_{i}\neq q_{i-1}\wedge L_{i}<N_{c},\;\mathrm{then}\;s\left[n\right]\leftarrow q_{i-1}\forall n\in \left[n_{i},n_{i+1}-1\right].$$
This procedure can be applied iteratively over $\left\{\left(q_{i-1},q_{i},q_{i+1}\right)\right\}$ until no violations remain, yielding a piecewise-constant phase trace without short-lived toggles and enabling stable inter-breath-interval and respiratory-rate estimation.

#### 3.4.2. Respiratory Rate Calculation

Respiratory rate is derived from the consolidated phase labels s[n] described previously. Figure 7 illustrates the outputs used here: the raw nostril-temperature sequence (green), its smoothed version (red), and the phase trace (orange) that marks inhalation, neutral/hold, and exhalation segments. Inter-breath intervals (IBI) are computed from consecutive phase transitions of a chosen event type (exhalation onsets in this implementation).

Let $f_{\mathrm{cam}}$ be the camera frame rate and define the exhalation-onset event set(52)$$E≜\left\{k|s[k-1]\neq +1\wedge s[k]=+1\right\}.$$
For consecutive events $k_{i-1},k_{i}\in E$ with $k_{i}>k_{i-1}$, the IBI (seconds) is(53)$$\Delta t_{i}=\frac{k_{i}-k_{i-1}}{f_{\mathrm{cam}}}.$$
Physiological plausibility is enforced consistently with the 5–42 BPM operating band by accepting only(54)$$\Delta t_{\min}\leq \Delta t_{i}\leq \Delta t_{\max},\;\Delta t_{\min}=\frac{60}{42}s,\;\Delta t_{\max}=\frac{60}{5}s.$$

Each validated interval yields an instantaneous respiratory rate(55)$$RR_{i}=\frac{60}{\Delta t_{i}}\left(\mathrm{BPM}\right).$$
To obtain a stable yet responsive trace as in Figure 7, two lightweight smoothers are applied sequentially: a causal weighted update(56)$${\tilde{RR}}_{i}=0.6RR_{i}+0.4{\tilde{RR}}_{i-1},\;{\tilde{RR}}_{0}=RR_{0},$$
followed by an exponential moving average (EMA) with coefficient α=0.7,(57)$$RR_{\mathrm{final},i}=\alpha {\tilde{RR}}_{i}+\left(1-\alpha \right)RR_{\mathrm{final},i-1},\;RR_{\mathrm{final},0}={\tilde{RR}}_{0}.$$
This IBI → weighted-average → EMA pipeline yields a robust BPM estimate under motion, thermal drift, and noise while remaining suitable for real-time embedded execution. For real-time visualization, a PyQt5 window renders the thermal video feed with ROI detection result overlays alongside the breathing waveform panel as well as the current breathing phase, respiratory rate in BPM, nostril temperature, ROI pixel area and the FPS of system as illustrates in Figure 8.

## 4. Experimental Results

### 4.1. Hardware and Software Configuration

The respiratory rate monitoring system was implemented on a Jetson Orin Nano Developer Kit (NVIDIA Corporation, Santa Clara, CA, USA; 6-core ARM Cortex-A78AE CPU, 8 GB LPDDR5 RAM) running Ubuntu 20.04 LTS. The YOLO-based nostril detection and respiratory-signal processing algorithms were developed in Python 3.10 with OpenCV 4.x and executed directly on the embedded device. Thermal video streams were processed in real-time, with inference and signal analysis performed entirely on the edge device, without requiring cloud-based computation. The raw temperature data were continuously collected from the TOPDON TC001 thermal camera (Topdon Technology Co., Ltd., Shenzhen, China) with a resolution of 256×192 pixels and a lightweight design (30 g). The camera was connected directly to the Jetson Orin Nano for on-device processing, ensuring minimal latency and maintaining a non-intrusive measurement environment.

To quantify the computational complexity of the proposed system on embedded hardware, the end-to-end execution was profiled over a 60 s continuous run on the Jetson Orin Nano. The average per-frame processing time was 65.2 ms, with a 95th-percentile latency of 85.3 ms, indicating that 95% of frames were processed within this bound. This corresponds to a real-time throughput of 22.5 FPS, with a standard deviation of 1.8 FPS, reflecting stable runtime characteristics throughout the measurement interval. As shown in Table 1, YOLO-based nostril detection constituted the primary computational load (35.5 ms, 54.5%), followed by thermal capture latency (15.0 ms, 23.3%) and graph/GUI updates (10.8 ms, 16.6%). All remaining modules, including temperature extraction (2.8 ms), adaptive phase detection and signal processing (1.5 ms), Kalman tracking (1.2 ms), and IBI calculation (0.5 ms), each contributed less than 5% of the total per-frame cost. System resource monitoring further showed moderate utilization, with mean CPU usage of 42.5%, GPU usage of 68.2%, and memory consumption of 850 MB. These results confirm that the computational burden is lightweight and well within the real-time operating envelope of low-power embedded edge devices.

### 4.2. Respiratory Rate Experimental Procedures

A pilot study was conducted with ten healthy adults (N = 10, aged 33.3 ± 4.38) under institutional ethics approval and written informed consent. Ground-truth RR was obtained by dual-rater manual counts from the recordings; for metronome-paced blocks, the target rate was logged as an auxiliary reference. To establish ground-truth RR, manual tally counting [51] was performed on the experimental video recordings. Each breathing cycle was visually identified by observing airflow from the nostril, and the total number of cycles within a predefined time window was recorded. The respiratory rate (RR) was then calculated as:(58)$$RR=\frac{N_{\mathrm{breaths}}}{T}\times 60\;\left[\mathrm{bpm}\right],$$
where $N_{\mathrm{breaths}}$ denotes the number of observed breathing cycles and *T* represents the duration of the observation in seconds. For example, if 22 breaths were observed during a 60-s video, the reference RR was 22 bpm.

The experimental protocol, summarized in Table 2 and the subject faced the thermal camera as illustrated in Figure 9, was designed not only to validate respiratory rate (RR) estimation under controlled conditions but also to emulate scenarios relevant to long-lie incidents. In a long-lie situation, an individual may remain immobile for an extended duration in various postures or under partial occlusions, where reliable respiration monitoring becomes a key indicator of consciousness and vitality. The resting and paced breathing sessions establish baseline accuracy across normal and rhythmic respiration patterns, forming the reference for physiological consistency. The robustness (soft speech) condition introduces mild facial motion to evaluate tolerance to articulation, representative of irregular speech or groaning that may occur before or after a fall. The distance and off-axis yaw conditions simulate variations in camera placement and subject orientation that would naturally arise when the person is lying at different angles or when the thermal sensor is mounted in a fixed overhead position. Finally, the posture (supine) recordings directly mimic a post-fall scenario, where the subject lies facing upward with minimal motion. Collectively, these conditions ensure that the proposed system is trained and validated under realistic variability, facilitating robust respiratory monitoring during long-lie detection.

### 4.3. Nostril Detection Performance

The nostril detection model was trained using a YOLO-based architecture, showing rapid and stable convergence, as evidenced by the steady decrease in box, classification, and distribution focal losses across both the training and validation sets, as illustrated in Figure 10. The evaluation metrics, including precision, recall, mAP@0.5, and mAP@0.5–0.95, exhibit consistent improvement and stabilization across epochs, confirming the robustness and generalization capability of the trained model for reliable nostril localization in thermal imagery.

The training curves demonstrate a steady increase in precision and recall, reaching over 99% within the first few epochs. Quantitative evaluation metrics in Figure 11 further confirm the model’s robustness, with the Precision–Recall (PR) curve showing an area under the curve (AUC) of 0.992, and the F1–Confidence curve peaking at 0.99. Both Precision–Confidence and Recall–Confidence curves indicate stable predictions across a wide confidence range, with optimal performance observed at a confidence threshold of approximately 0.86.

To complement the quantitative analysis, Figure 12 provides an enlarged view of the nostril detection output, making the detection label and bounding box clearly visible. Meanwhile, Figure 13 presents qualitative examples of the YOLO-based nostril detector applied to diverse thermal video frames. These examples demonstrate consistent and reliable nostril localization under various conditions, including different head poses, lighting variations, and partial occlusions. The trained model accurately identifies the nostril region across diverse thermal video frames, demonstrating robustness and stability for downstream respiratory rate estimation tasks.

### 4.4. Respiratory Rate Estimation Accuracy

Before estimating the respiratory rate following the procedures in Section 4.2, the nostril detector was first validated to ensure reliable localization across different subject poses, as illustrated in Figure 14, showing the thermal frame was collected and aligned with the experimental setup in Table 2. The automatically detected nostril regions (green bounding boxes) are shown across a wide range of conditions, including resting, metronome-paced breathing, soft-speech influence, distance variation, off-axis head orientation, and posture changes. This variability ensures that the evaluation reflects realistic operating scenarios with differences in viewpoint, articulation, and body orientation.

Once the nostril ROI is successfully detected, the system extracts the corresponding temperature signal to determine the breathing phases. Figure 15 presents representative nostril-temperature waveforms under four typical experimental conditions: resting, paced breathing (24 BPM), soft speech, and off-axis yaw. The green line indicates the raw temperature sequence, the red line depicts the smoothed and band-pass-filtered signal, and the orange-shaded regions denote exhalation phases identified by the adaptive MAD–hysteresis algorithm. During resting, as shown in Figure 15a, the thermal oscillations are smooth and periodic, reflecting stable nasal airflow. Under paced breathing in Figure 15b, the oscillation frequency increases in line with the metronome rhythm, confirming temporal consistency with the ground-truth reference. In soft-speech (Figure 15c) and off-axis (Figure 15d) scenarios, irregularities appear due to motion and partial ROI displacement; however, the system still successfully tracks the phase transitions, demonstrating robustness to moderate motion and physiological variability.

The respiratory-rate estimation experiment was conducted with ten healthy participants (age: 33.3 ± 4.38 years). Each subject completed six breathing conditions described in Table 2, performed in real-time under different room layouts and lighting environments.

A summary of the results is presented in Table 3, which provides a per-subject breakdown of MAE, RMSE, and ROI pixel area across the six conditions, highlighting both inter-subject and condition-specific variability. Most participants exhibit stable performance during resting and paced breathing, whereas higher errors emerge during soft-speech and distance-related conditions due to motion artifacts and reduced signal-to-noise ratio (SNR). Notably, although the ROI area during soft speech remains larger than that in the distance condition, the estimation error is still considerably higher, suggesting that dynamic facial motion, rather than ROI size, is the primary factor contributing to performance degradation.

The correlation between the estimated and reference respiratory rates is shown in Figure 16a, demonstrating a strong linear relationship with $R^{2}=0.973$. These results indicate that the system can reliably estimate respiratory rate with minimal error across different conditions and sessions. To further examine the condition-wise performance, Figure 16b compares the distribution of estimated and reference respiratory rates using box plots, revealing close alignment across most conditions, with only slight deviations observed during soft-speech and off-axis orientations.

Moreover, Figure 16c illustrates the per-subject MAE distribution across the six conditions, highlighting both individual variability and condition-dependent performance. Estimation remains stable during resting and paced breathing, whereas larger errors appear during soft speech, off-axis yaw, and increased camera distance due to motion-induced disturbances and weakened thermal signal fidelity. Figure 16d summarizes the average MAE, RMSE, and ROI size per condition, showing that accuracy declines beyond 1.5 m as the nostril region becomes smaller and the thermal contrast diminishes. Interestingly, despite having a larger ROI area than the distance condition, the soft-speech condition exhibits higher estimation errors, reinforcing that facial dynamics, rather than ROI scale, are the dominant factor affecting accuracy.

Table 4 summarizes the average respiratory-rate estimation performance across all tested conditions, expressed as mean ± SD for both MAE and RMSE. The results demonstrate that the system achieves consistently low errors across all scenarios, with the lowest MAE and RMSE observed during resting and paced breathing. Under conditions involving speech or posture change, the estimation error slightly increases, reflecting temperature fluctuations and ROI variation. For comparison, the included peak-based and FFT-based baseline methods show substantially higher errors across all conditions, confirming the advantage of the proposed adaptive approach. The overall error remains below 1 BPM on average, confirming clinically acceptable [52] performance for a lightweight thermal-based system operating on an embedded device.

To further investigate the influence of distance on detection scale and estimation accuracy, Table 5 reports the mean ± SD of MAE, RMSE, and ROI size across the three measurement distances. The results show a substantial reduction in detected ROI area as the camera moves farther away—from 597 px^2^ at 1.0 m to just 165 px^2^ at 2.0 m—representing a 72% decrease in spatial sampling. This loss of pixel coverage directly diminishes thermal contrast and reduces signal amplitude, leading to higher estimation errors at extended distances. The increasing error trend therefore, aligns with the shrinking ROI size, confirming that reduced spatial resolution limits the system’s ability to capture subtle nostril temperature variations. Nevertheless, within the practical monitoring range of 1.0–1.5 m, estimation performance remains stable, with MAE values below 0.7 BPM.

A more detailed visualization of the distance–accuracy relationship is provided in Figure 17. As shown in Figure 17a, the detection error increases markedly, from 0.27 BPM MAE and 0.31 BPM RMSE at 1 m to 1.38 BPM MAE and 1.52 BPM RMSE at 2 m, while the corresponding ROI area decreases from approximately 597 px^2^ to 165 px^2^ (Figure 17b). This consistent trend reinforces that distance-induced loss of spatial detail is the primary factor driving performance degradation.

To contextualize these device-level gains, the proposed system is compared with recent contactless respiratory-rate estimation studies. Table 6 summarizes relevant methods using thermal imaging, highlighting differences in hardware, ROI selection, tracking strategy, estimation algorithm, accuracy, runtime feasibility, and overall contribution. The developed system achieves markedly lower estimation errors than most recent thermal-based approaches, attaining a mean absolute error of 0.57 ± 0.36 BPM and an RMSE of 0.64 ± 0.42 BPM across diverse conditions, including speech, head rotation, and distances up to 2.0 m. These results fall well within the commonly accepted clinical tolerance for respiratory-rate monitoring (error < 2 BPM) [52], corresponding to an average deviation below 1 BPM. The system thus delivers clinically relevant accuracy using a lightweight 256×192 thermal camera.

Compared to deep learning–based approaches, which achieve good accuracy but require high-resolution cameras, complex models, and non-real-time post-processing [19,26], the proposed system emphasizes lightweight edge deployment with minimal computational cost. Cross-modality solutions that fuse RGB and thermal data have demonstrated clinical viability but introduce additional sensor complexity and are not optimized for embedded platforms [22]. Meanwhile, traditional spectral methods achieve competitive accuracy but typically lack automated ROI localization and real-time performance [52,53,54]. In contrast, the proposed system is the first to integrate nostril-specific ROI tracking, adaptive MAD–hysteresis phase detection, and IBI validation within a real-time, edge-deployable framework. This combination enables robustness to motion and viewpoint variation while achieving state-of-the-art accuracy and real-time operation suitable for continuous home monitoring.

## 5. Discussion

The experimental results demonstrate that the proposed thermal-based system, which integrates a YOLO-based nostril detector executed on every second frame with Kalman prediction and an adaptive breathing-phase and IBI validation module, enables accurate and robust respiratory-rate estimation entirely on an embedded edge device. The frame-skipping strategy effectively reduces computational demand without compromising ROI continuity, while the time-domain phase logic, based on median and MAD thresholds with hysteresis and short-segment consolidation, maintains stable breathing-phase labeling under thermal drift and motion-induced noise. To strengthen the evaluation, two baseline methods, such as peak detection and FFT-based spectral analysis, were implemented and tested on the same real-time data collected in the study. As shown in Table 4, these baselines exhibit substantially higher MAE and RMSE across all conditions, confirming the advantage of the proposed method. Although several recently published thermal-based respiratory-rate estimation methods exist, direct comparison was not feasible. Most rely on substantially higher-resolution thermal sensors, computationally intensive 3D-CNN or transformer architectures, or RGB–thermal fusion pipelines that cannot be reproduced on the low-resolution dataset used in this study or deployed on embedded hardware.

In terms of privacy, the thermal modality used here does not capture facial texture, identity cues, or personally identifiable imagery. The 256×192 thermal frames contain only coarse temperature gradients, and the respiratory pipeline operates exclusively on a small nostril-level ROI, further reducing the possibility of re-identification. Although a formal privacy-impact assessment was not mandated for this pilot study, the sensing modality is inherently privacy-preserving compared with RGB-based approaches. When contrasted with genuinely anonymous alternatives such as radar or acoustic sensing, thermal imaging provides a favorable balance between privacy and spatial specificity: radar and acoustic systems offer strong anonymity but often exhibit reduced spatial precision, susceptibility to multipath or ambient noise, and difficulty maintaining stable anatomical anchoring for breath extraction [55,56]. By leveraging non-textured thermal data while retaining reliable nostril localization, the proposed approach achieves privacy-aware respiratory monitoring without compromising estimation accuracy.

Across all evaluated conditions, respiratory-rate estimation remained consistently accurate, with MAE values ranging from 0.34 to 0.98 BPM and RMSE values between 0.36 to 1.07 BPM across ten participants with an overall average of 0.57±0.36 BPM with RMSE 0.64±0.42 BPM. Errors were lowest during resting and paced-breathing trials, and increased modestly under soft-speech and posture variations due to mouth motion and partial ROI displacement. Distance and off-axis tests showed moderate increases in error, indicating that the system maintains reliable estimation up to approximately 1.5 m, even with a low-resolution thermal camera sensor. These results confirm that accurate, real-time respiratory-rate monitoring can be achieved using a adaptive, privacy-preserving thermal camera operating entirely on an edge platform. The experimental findings also demonstrate the adaptive nature of the proposed system across diverse conditions. Rather than relying on fixed thresholds or static parameters, the MAD-based phase logic continuously adjusts to variations in signal amplitude and noise, while the hysteresis and consolidation mechanisms ensure stable breathing-phase transitions. These adaptive behaviors collectively enable consistent performance under different breathing patterns and motion scenarios without manual recalibration.

A closer examination of failure modes provides further insight into the system’s behavior under challenging scenarios. During soft-speech condition, the primary source of degradation arose not only general facial motion but also from rapid upper-lip deformation and transient nostril occlusion caused by articulation. These movements introduce abrupt thermal discontinuities within the ROI, reducing local temporal coherence and lowering the effective signal-to-noise ratio (SNR) of the extracted temperature waveform. A similar failure trend is observed with increasing camera distance, where the nostril region shrinks from 597 px^2^ at 1.0 m to 165 px^2^ at 2.0 m. This reduction leads to diminished thermal gradient resolution, smaller oscillation amplitudes, and greater sensitivity to pixel-level quantization noise. Together, these analyses clarify the mechanisms underlying soft-speech- and distance-related performance degradation, complementing the quantitative results in Table 4.

Compared with existing methods summarized in Table 6, the primary advantage of the proposed system lies in its fully automated, real-time processing at minimal computational cost. Solely deep learning-based approaches can achieve strong performance but typically depend on high-resolution cameras and computationally intensive models, often requiring offline post-processing [19,26]. In contrast, conventional spectral methods can run on simpler hardware yet usually lack automated ROI localization and are sensitive to motion and baseline temperature drift [52,53]. The present design achieves both efficiency and stability by performing YOLO detections on thermal imagery, transferring the ROI via calibrated alignment, applying Kalman prediction on skipped frames, and estimating the breathing pattern directly in the time domain. This architecture maintains robustness to motion and noise while remaining fully compatible with real-time embedded execution.

A direct comparison with recent studies further contextualizes system performance. Mozafari et al. [26] reported an MAE of approximately 1.6 BPM using a 640 × 480 thermal camera with a 3D-CNN + BiLSTM model, while Nakai et al. [54] reported MAE values around 2.4 BPM using dual thermal ROIs. Gioia et al. [19] achieved an $R^{2}$ of roughly 0.10 with high-resolution imagery and offline 3D-CNN regression. Classical FFT/CZT-based approaches typically achieve MAE between 0.66 and 1.8 BPM under controlled conditions but depend on manual or semi-automated ROI selection [22]. In contrast, the proposed system delivers competitive accuracy using a lower-resolution 256×192 sensor while maintaining fully automated ROI tracking and real-time embedded execution. Furthermore, most prior thermal-based studies evaluated only one or two controlled breathing conditions, whereas the proposed system was validated across six diverse scenarios, including speech, off-axis rotation, posture variation, and distances up to 2.0 m, demonstrating robustness under a wider range of real-world variations.

Regarding model validity, the respiratory-rate estimation module does not involve any data-driven training and therefore cannot overfit to the subjects. All processing parameters, including band-pass filter settings, MAD-based thresholds, and hysteresis rules, were fixed in advance and applied identically to all participants. The YOLO nostril detector, the only trained component in the pipeline, was trained on an independent thermal dataset (7958 annotated frames) that did not include any of the subjects used in the RR evaluation. Ground-truth RR was obtained through dual-rater manual counting, and system accuracy was assessed using MAE and RMSE across all six experimental conditions.

Despite the advantages of the proposed method, several limitations were identified. The system requires a clearly visible nostril region to estimate the respiratory rate accurately. When the nostrils are covered, exhibit low thermal contrast, or move outside the camera’s field of view, such as when subjects wear masks or turn their heads excessively, the system may fail to produce valid respiratory rate estimates because no stable thermal signal can be extracted. In addition, accuracy decreases as the camera-to-subject distance increases. Beyond approximately 1.5–2.0 m, the nostril region becomes very small in the thermal frame, the average ROI area decreases from about 597 px^2^ to 165 px^2^, reducing temperature contrast and making the signal more susceptible to noise and minor tracking errors. Moreover, the proposed method is effective only when breathing occurs predominantly through the nasal pathway. During diaphragmatic or abdominal breathing, where nasal airflow is minimal, the thermal contrast around the nostrils becomes negligible, leading to weak or undetectable respiratory oscillations. Furthermore, when the ambient temperature is high, the subject’s facial temperature increases, diminishing the thermal contrast and making face or nostril detection unreliable.

As this work represents a feasibility study to demonstrate whether a low-resolution thermal camera can reliably provide non-invasive respiratory-rate monitoring in real time, the evaluation was intentionally limited to ten healthy adults within a narrow age range (mean age 33.3 ± 4.38 years). Consequently, elderly adults or individuals with respiratory conditions such as COPD, asthma, or sleep apnoea were not included, resulting in limited clinical validation and reduced population diversity. Future work will conduct broader clinical validation involving these patient groups, as well as individuals at risk of long-lie incidents, to ensure that the system performs reliably across diverse real-world populations.

To support these clinical applications, the system also requires further technical enhancements to ensure reliable respiration monitoring under more challenging real-world conditions. Future improvements may include integrating a low-power radar module to complement the thermal sensing, enabling respiration-phase estimation even when nasal airflow is weak or partially occluded. Beyond nasal-based measurements, future work will also explore extracting respiratory information from micro-motions of the shoulder or abdomen and identifying mouth-breathing episodes. Furthermore, fusing the proposed respiratory-rate estimator with the previously developed long-lie detection system [57] may improve reliability and reduce false alarms across varying distances, postures, and environmental conditions. This multi-modal integration would move the system toward more robust and clinically relevant continuous home monitoring.

## 6. Conclusions

This paper introduces an adaptive, fully automated, and privacy-preserving respiratory-rate monitoring system based on thermal imaging, designed for real-time execution on embedded edge hardware. The framework integrates a lightweight thermal-specific YOLO-based nostril detector, a detector-centric frame-skipping strategy with Kalman prediction for stable ROI continuity, and an adaptive median–MAD hysteresis algorithm with consolidation and IBI validation for robust time-domain respiration analysis.

Across six experimental conditions, including speech, off-axis rotation, posture variation, and distances up to 2.0 m, the system achieved an average MAE of 0.57±0.36 BPM and RMSE of 0.64±0.42 BPM, demonstrating that accurate and reliable respiratory-rate estimation is achievable using a compact 256×192 thermal sensor operating fully on a low-power embedded platform. The adaptive signal-processing pipeline consistently adjusted to variations in breathing amplitude, rhythm, and motion-induced disturbances without requiring manual recalibration. Notably, the achieved accuracy falls well within clinically acceptable tolerances for respiratory-rate monitoring, reinforcing its suitability for practical deployment in home or long-term monitoring environments.

Future work will include expanding the participant population and range of respiratory scenarios, improving resilience against occlusion and abdomen- or mouth-dominant breathing through the integration of a low-power radar module, and embedding the proposed respiratory module into an existing long-lie detection system. This multi-modal fusion of thermal and physiological information aims to enhance robustness, reduce false alarms, and support continuous, privacy-preserving home monitoring.

## Figures and Tables

**Figure 1 sensors-26-00278-f001:**
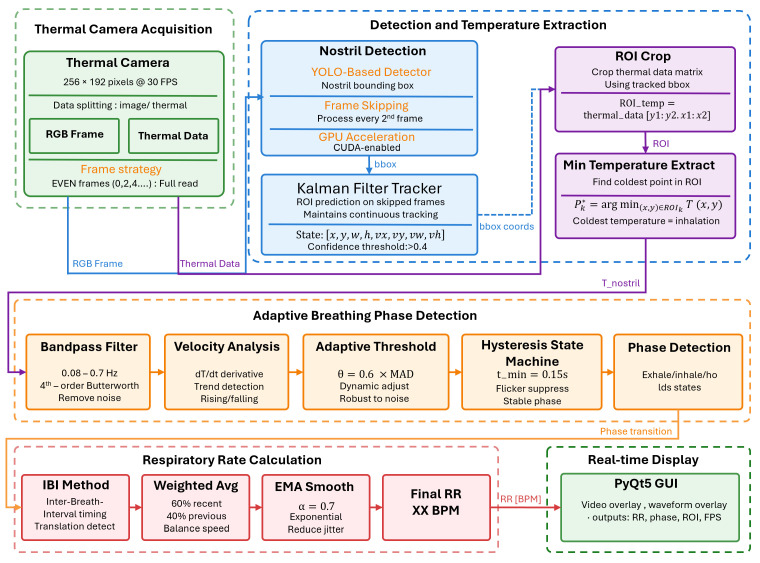
System overview of the proposed real-time thermal-based respiratory rate estimation. YOLO-based detector with Kalman tracking stabilizes the nostril region of interest (ROI), from which the coldest pixel temperature is extracted as the airflow-related signal. ROI min-temperature is band-pass filtered and analysed with MAD-based hysteresis and IBI validation to produce respiratory rate.

**Figure 2 sensors-26-00278-f002:**
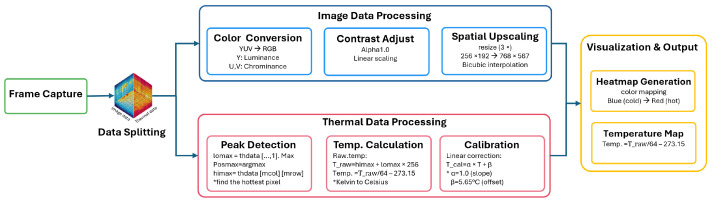
Processing pipeline of thermal camera acquisition. The raw frame is split into two streams: image data (top) for visualization and heatmap generation, and thermal data (bottom) for temperature calculation, calibration, and ROI detection. This split is specific to the TOPDON TC001 dual-stream frame format. The asterisk (*) indicates that each block is further explained in the subsequent text to emphasize its specific role within the pipeline.

**Figure 3 sensors-26-00278-f003:**
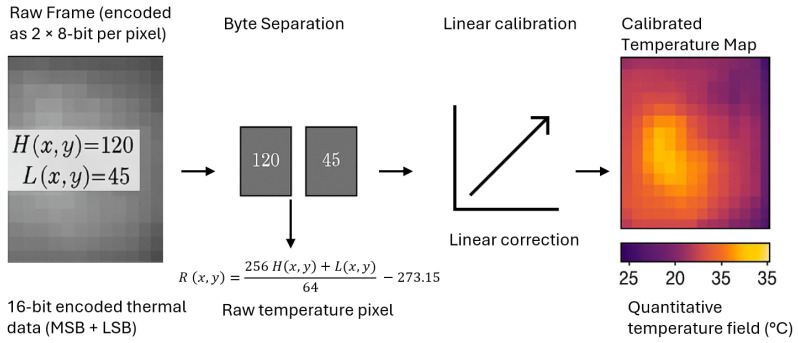
Thermal data decoding and calibration process. Each pixel is formed from two 8-bit bytes (MSB + LSB) combined to reconstruct R(x,y), followed by linear calibration to produce the quantitative temperature map (°C).

**Figure 4 sensors-26-00278-f004:**
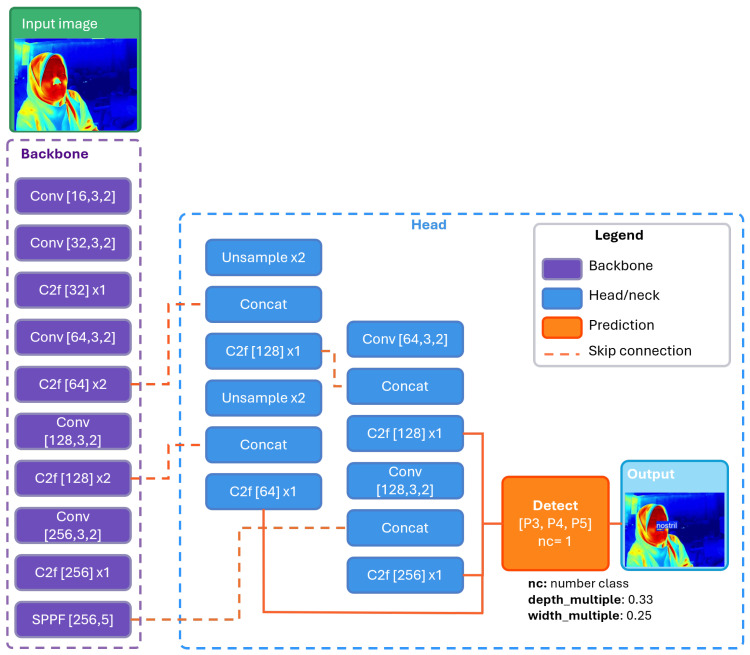
Architecture of the YOLO-based nostril detection model for thermal imagery. The backbone extracts multi-scale thermal features, while the head performs feature fusion and detection across P3–P5 layers to localize the nostril region in real time. The blue dashed outline indicates the boundary of the detection head, and overlapping elements are used for compact visualization without affecting architectural interpretation.

**Figure 5 sensors-26-00278-f005:**
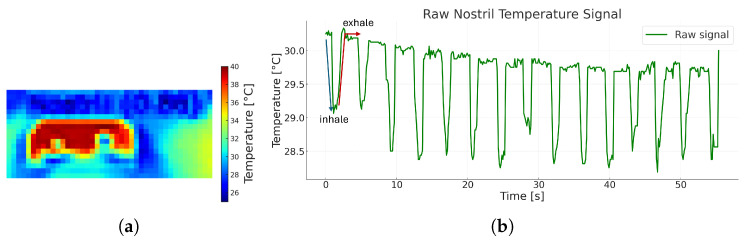
Illustration of respiratory signal extraction. (**a**) Cropped nostril ROI represented in the thermal domain; (**b**) corresponding raw nostril temperature signal S, where the oscillations of ${\hat{T}}_{k}$ denote inhalation (cooling) and exhalation (warming), indicated by the blue and red lines, respectively, providing the input for subsequent signal processing and respiratory rate estimation.

**Figure 6 sensors-26-00278-f006:**
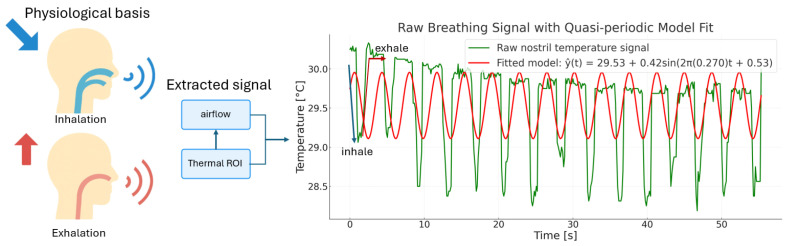
Schematic illustration of the proposed respiration monitoring framework. Inhalation of cooler air lowers nostril temperature, while exhalation of warmer air raises it (**left**), illustrated by blue and red arrows, respectively. The airflow-induced modulation is captured from the thermal ROI and converted into a raw nostril temperature signal (**middle**). The extracted signal (green) exhibits quasi-periodic fluctuations that can be approximated by a sinusoidal model (red), reflecting the underlying breathing cycles (**right**).

**Figure 7 sensors-26-00278-f007:**
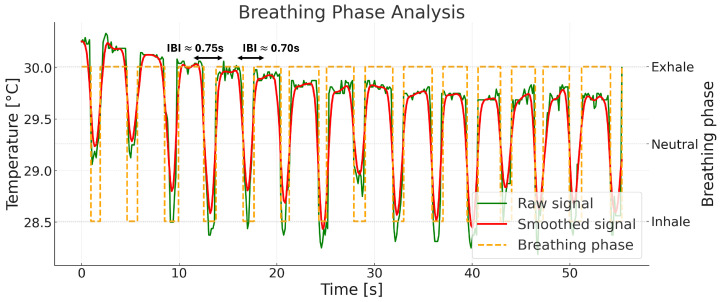
Breathing phase detection from nostril temperature signals. Raw (green) and smoothed (red) sequences are shown, with detected breathing phases indicated by orange dashed lines corresponding to exhalation phases; inhalation and hold are implicitly identified by phase transitions.

**Figure 8 sensors-26-00278-f008:**
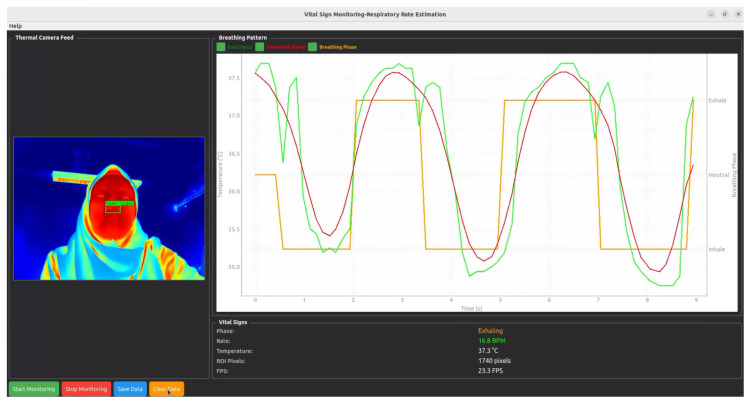
Graphical user interface (GUI) of the proposed respiratory rate monitoring system. The left panel shows the thermal camera feed with automated nostril detection. The right panel illustrates the extracted nostril temperature signal, the green, red, and orange lines denote the raw signal, smoothed signal, and detected breathing phase, respectively. The bottom panel provides real-time vital signs, including current breathing phase, respiratory rate in BPM, nostril temperature, and processing frame rate (FPS).

**Figure 9 sensors-26-00278-f009:**
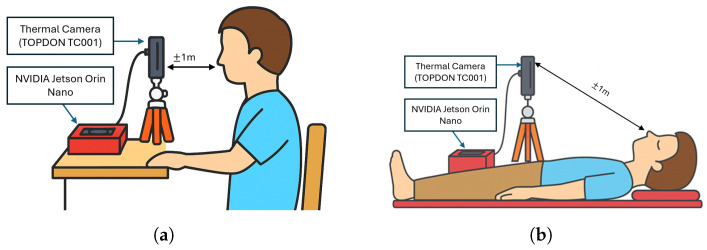
Illustration of experimental setup. (**a**) Seated configuration: thermal camera aligned at nose height at 1–2 m to capture nostril temperature; (**b**) Supine configuration: camera pitched downward to preserve nasal visibility.

**Figure 10 sensors-26-00278-f010:**
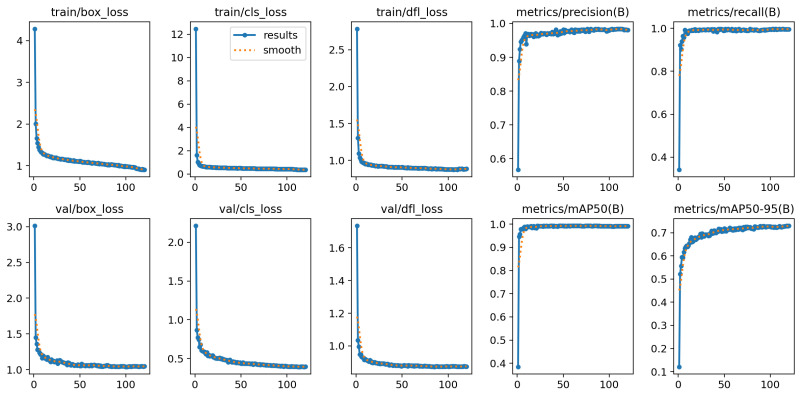
Training performance curves of the YOLO-based nostril detection model. The plots show the evolution of box loss, classification loss, distribution focal loss (DFL), and evaluation metrics (precision, recall, mAP@0.5, and mAP@0.5–0.95) for both training and validation datasets over the training epochs.

**Figure 11 sensors-26-00278-f011:**
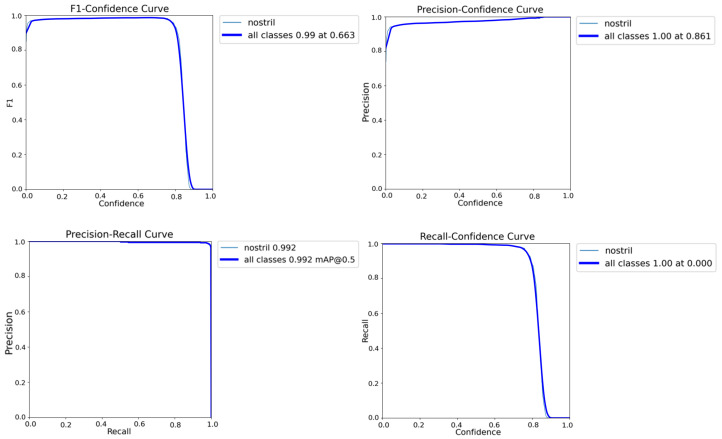
Performance curves of the YOLO-based nostril detection model on the validation dataset including F1–Confidence curve, Precision–Confidence curve, Precision–Recall (PR) curve, and Recall–Confidence curve. The results indicate consistently high precision, recall, and F1-scores across a wide confidence range, with optimal detection performance achieved at confidence thresholds between 0.66 and 0.86.

**Figure 12 sensors-26-00278-f012:**
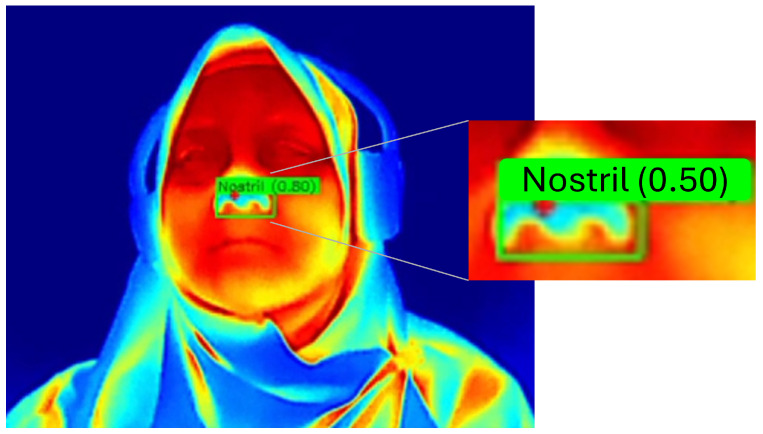
Close-up example of the detected nostril region. The enlarged crop highlights the detection label and bounding box that are less visible in the full-resolution montage presented in Figure 13.

**Figure 13 sensors-26-00278-f013:**
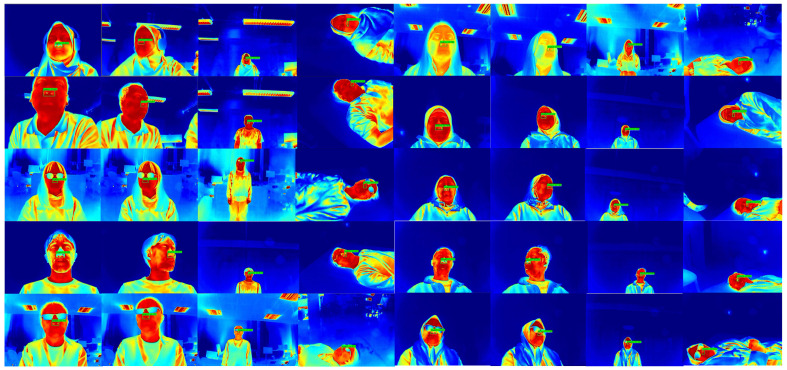
Qualitative results of the YOLO-based nostril detection model on thermal video frames. The model consistently localizes the nostril region with high confidence across varying head poses, facial orientations, and partial occlusions.

**Figure 14 sensors-26-00278-f014:**
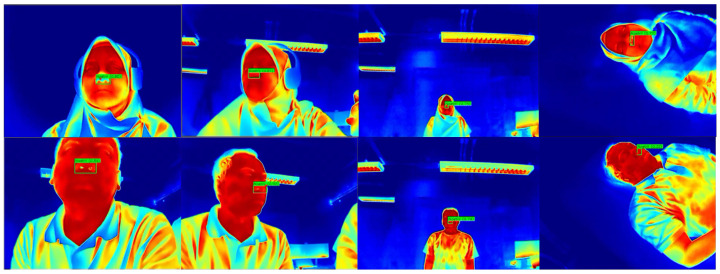
Representative thermal frames showing automatically detected nostril regions (green bounding boxes) across different participants and experimental conditions, including relaxed, off-axis, distance (2 m) and supine.

**Figure 15 sensors-26-00278-f015:**
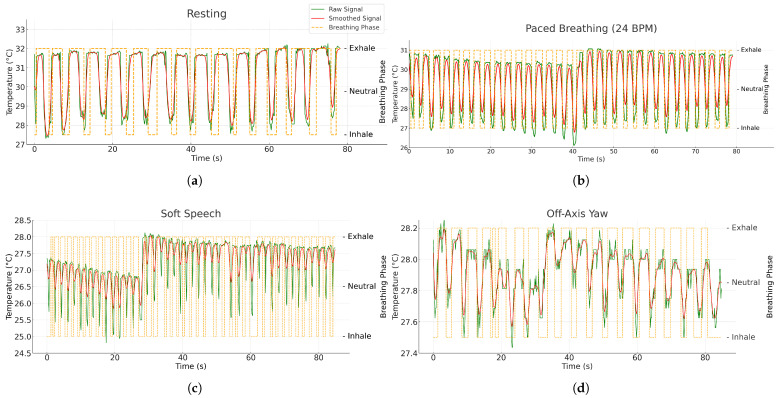
Representative breathing pattern signals extracted from the nostril ROI across four experimental conditions: (**a**) resting, (**b**) paced breathing, (**c**) soft speech, (**d**) off-axis yaw.

**Figure 16 sensors-26-00278-f016:**
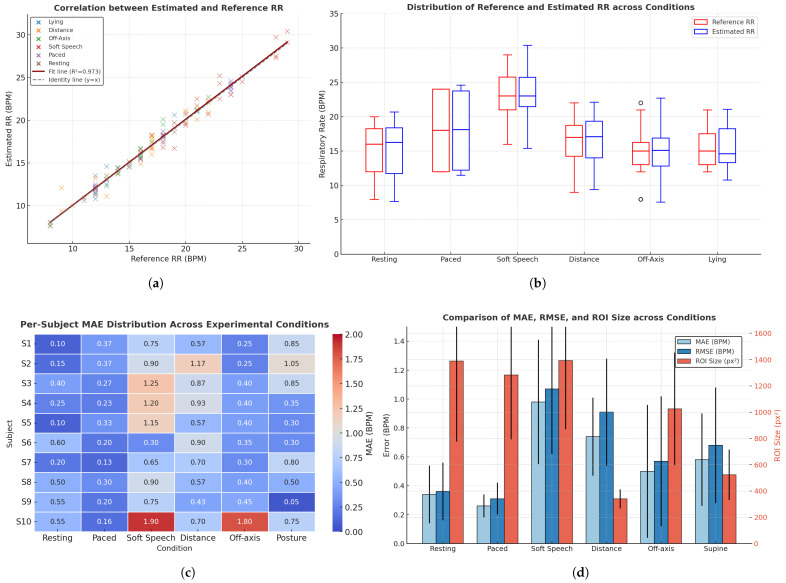
Summary of respiratory rate estimation results across all experimental conditions. (**a**) Correlation between estimated and reference respiratory rates, showing strong agreement across all breathing conditions. (**b**) Box-plot comparison between reference and estimated respiratory rates under different conditions, illustrating close alignment and low dispersion. (**c**) Per-subject MAE distribution across six experimental conditions, highlighting individual and condition-specific variability. (**d**) Overall MAE, RMSE, and ROI-size analysis per condition, showing accuracy degradation as camera distance increases due to reduced nostril-region pixels and weakened thermal contrast; notably, during soft speech, increased facial motion leads to higher estimation error despite a larger ROI area compared with the distance condition.

**Figure 17 sensors-26-00278-f017:**
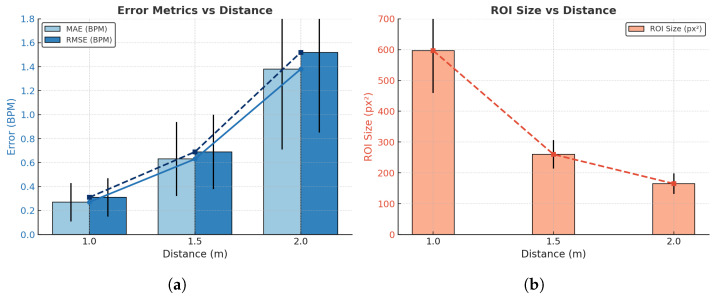
The effect of measurement distance on respiratory rate estimation performance. (**a**) shows the variation of error metrics (MAE and RMSE) with increasing distance, where solid bars denote MAE and dashed lines denote RMSE; (**b**) presents the corresponding change in ROI size across different measurement distances, where the dashed line highlights the overall trend.

**Table 1 sensors-26-00278-t001:** Profiling summary of computational performance on the Jetson Orin Nano (60 s run).

(A) Runtime Breakdown per Component					
Component		Mean (ms)		p95 (ms)	Percentage
Total Frame Time		65.20		85.30	100%
YOLO Detection		35.50		42.10	54.5%
Thermal Capture		15.00		30.00	23.3%
Graph/GUI Update		10.80		14.20	16.6%
Signal Processing		3.50		3.80	5.4%
Temperature Extraction		2.80		3.50	4.3%
Kalman Tracking		1.20		0.80	1.8%
IBI Calculation		0.50		0.80	0.8%
**(B) System Resource Utilization**				**(C) End-to-End Throughput Statistics**	
Metric	Mean	p05	p95	Metric	Value (FPS)
CPU Usage	42.5%	35.2%	58.3%	Mean FPS	22.5
GPU Usage	68.2%	67.5%	80.3%	Min FPS	18.2
Memory Usage	850.3 MB	820.5 MB	890.2 MB	Max FPS	28.8
				p50	24.5
				p95	24.5
				Std Dev	1.8

**Table 2 sensors-26-00278-t002:** Experimental design overview for respiratory rate data collection. Each block is a 60 s recording.

Condition Set	Blocks per Subject	Duration	Description
Resting (spontaneous)	2×	60 s	Seated, natural nasal breathing, mouth closed
Paced breathing (metronome)	3×	60 s	Seated; guided at 12, 18, 24 BPM (randomized order); metronome target logged as auxiliary reference
Robustness (soft speech)	2×	60 s	Seated; counting aloud to emulate mild articulatory motion
Distance (stood)	3×	60 s	Spontaneous breathing at 1.0, 1.5, and 2.0 m; camera height/pitch held constant
Off-axis yaw (seated)	2×	60 s	Spontaneous breathing at $\pm 30^{\circ }$ yaw; neutral pitch/roll instructed
Posture (supine)	2×	60 s	Spontaneous breathing in supine facing camera; camera pitched downward

**Table 3 sensors-26-00278-t003:** MAE, RMSE, and ROI pixel area (px^2^) of respiratory rate estimation per subject under six experimental conditions (pilot evaluation, *N* = 10).

Subj.	Resting			Paced			Soft Speech			Distance			Off-Axis			Posture		
	MAE	RMSE	px^2^	MAE	RMSE	px^2^	MAE	RMSE	px^2^	MAE	RMSE	px^2^	MAE	RMSE	px^2^	MAE	RMSE	px^2^
S1	0.1	0.1	2254	0.37	0.4	1847	0.75	0.99	1779	0.57	0.74	364	0.25	0.35	1216	0.85	1.33	808
S2	0.15	0.21	1742	0.37	0.46	2045	0.9	0.98	1888	1.17	1.81	493	0.25	0.29	1420	1.05	1.18	769
S3	0.4	0.41	809	0.27	0.32	785	1.25	1.57	709	0.87	1.14	286	0.4	0.5	831.5	0.85	0.91	544
S4	0.25	0.25	1390	0.23	0.35	1065	1.2	1.3	1528	0.93	1.02	367	0.4	0.5	831	0.35	0.38	500
S5	0.1	0.1	1401	0.33	0.42	1359	1.15	1.16	1984	0.57	0.70	273	0.4	0.41	1164	0.3	0.36	341
S6	0.6	0.6	2359	0.2	0.22	1738	0.3	0.36	1705	0.90	0.95	329	0.35	0.49	1897	0.3	0.31	597
S7	0.2	0.28	1102	0.13	0.14	846	0.65	0.65	861	0.70	0.79	261	0.3	0.42	777	0.8	1.06	301
S8	0.5	0.59	713	0.3	0.34	624	0.9	0.95	731	0.57	0.68	360	0.4	0.41	542	0.5	0.54	427
S9	0.55	0.57	1081	0.2	0.29	975	0.75	0.79	1046	0.43	0.55	317	0.45	0.51	935	0.05	0.07	496
S10	0.55	0.57	1055	0.16	0.17	1559	1.9	1.94	1696	0.70	0.72	352	1.8	1.82	644	0.75	0.87	446

**Table 4 sensors-26-00278-t004:** Respiratory rate estimation errors across all experimental conditions (mean ± SD, N=10). Baseline peak/FFT methods use one representative trial per subject due to their high sensitivity to artifacts, while the proposed method uses all recordings.

Condition	MAE (BPM)			RMSE (BPM)			ROI (px^2^)
	Proposed	Peak	FFT	Proposed	Peak	FFT	
Resting (Spontaneous)	0.34 ± 0.20	10.07 ± 5.48	1.51 ± 1.30	0.36 ± 0.20	11.24 ± 5.48	2.10 ± 1.30	1390 ± 614
Paced Breathing (Metronome)	0.26 ± 0.08	0.93 ± 0.65	5.11 ± 5.90	0.31 ± 0.11	1.12 ± 0.65	7.58 ± 5.90	1284 ± 491
Robustness (Soft Speech)	0.98 ± 0.43	4.39 ± 2.99	3.70 ± 3.92	1.07 ± 0.45	5.22 ± 2.99	5.24 ± 3.92	1393 ± 522
Distance (1.0–2.0 m)	0.74 ± 0.27	8.33 ± 5.19	5.76 ± 5.77	0.91 ± 0.37	9.68 ± 5.19	7.94 ± 5.78	340 ± 206
Off-axis Yaw ($\pm 30^{\circ }$)	0.50 ± 0.46	10.72 ± 3.80	2.31 ± 3.79	0.57 ± 0.45	11.30 ± 3.80	3.78 ± 3.79	1026 ± 428
Posture (Supine)	0.58 ± 0.32	6.31 ± 4.04	3.47 ± 3.94	0.68 ± 0.40	7.38 ± 4.04	5.10 ± 3.94	523 ± 192
Overall	0.57 ± 0.36	6.79 ± 3.86	3.64 ± 3.88	0.64 ± 0.42	7.58 ± 3.86	5.48 ± 3.88	–

**Table 5 sensors-26-00278-t005:** Effect of distance on respiratory rate estimation accuracy and ROI size (mean ± SD, *N* = 10).

Distance (m)	MAE (BPM)	RMSE (BPM)	ROI Size (px^2^)	Observation *
1.0	0.27 ± 0.16	0.31 ± 0.16	597 ± 138	Clear nostril region, distinct thermal contrast
1.5	0.63 ± 0.31	0.69 ± 0.31	260 ± 46	Reduced contrast, smaller ROI
2.0	1.38 ± 0.67	1.52 ± 0.67	165 ± 33	Weak contrast, partial pixel loss

* Observations are based on visual inspection of thermal contrast and ROI clarity during recordings.

**Table 6 sensors-26-00278-t006:** Comparative summary of recent contactless respiratory rate (RR) estimation methods using thermal imaging. Each method is evaluated based on subjects/conditions, camera specifications, setup (including ROI and method), accuracy, runtime feasibility, and novelty. The proposed system emphasizes nostril-focused ROI, IBI validation, and real-time edge deployment.

Study	Subjects & Conditions	Camera Spec & Setup (ROI + Method)	Accuracy (BPM)	Real-Time	Contribution
Ours	10 adults; six 60-s sets (resting, paced 12/18/24 BPM, soft speech, distance 1–2 m, yaw ± 30°, posture supine)	TOPDON TC001 (256 × 192); RO: Ithermal YOLOv8n (even frames, s = 10)+ Kalman tracking on skipped frames; band-pass 0.08–0.7 Hz; adaptive MAD–hysteresis phase detection + IBI validation	MAE (mean 0.57±0.36); RMSE (mean 0.64±0.42)	Yes	Thermal-based YOLO detector with Kalman tracking; adaptive MAD–hysteresis phase and IBI validation
Gioia et al. [19]	30 adults; 5-min tasks (rest, Stroop, emotion); RR 9–30	FLIR T640 (640 × 480); ROI: upper lip/nose (manual); 3D-CNN end-to-end regression	$R^{2}\approx 0.61$ (no MAE/RMSE)	No	Feasibility of end-to-end deep learning directly from thermal video
Mozafari et al. [26]	22 adults; sitting/standing × mask/no-mask, 90 s	FLIR T650sc (640×480); ROI: full face (DeTr); 3D-CNN + BiLSTM with correlation loss	MAE 1.6±0.4	Yes	Deep learning robust to mask & posture; real-time feasibility focus
Maurya et al. [22]	14 adults (rest, talking, variable); 8 neonates (NICU)	FLIR-E60 (320 × 240) + Logitech C922 RGB (960 × 720); ROI: nose–mouth (from RGB mapped to thermal); Hampel+MA+BP filtering; CZT spectral analysis	Adults: MAE 0.10–1.8; Neonates: MAE ≈ 1.5	No	Cross-modality ROI mapping; validated adults & neonates
Takahashi et al. [52]	7 adults; paced 15–30 BPM	FLIR Boson 320 (320 × 256); ROI: face subregions (scored by RQI); YOLOv3 + RQI; FFT on best region	MAE 0.66; LoA ± 2 BPM	No	ROI quality index (RQI) for automated subregion selection
Pereira et al. [53]	12 adults (rest, pathological); 8 neonates (NICU)	InfraTec VarioCAM HD (1024 × 768); ROI: full face (multi-grid, black-box); adaptive spectral analysis (autocorr, AMDF, peak detection)	RMSE 0.31 (rest), 3.27 (varied), 4.15 (neonates)	No	First NICU validation; black-box ROI without anatomical landmark
Nakai et al. [54]	11 healthy adults; seated at 1 m distance (lab environment)	FLIR A315 (320 × 240, 60 fps); manually defined nose & shoulder ROIs; dual-signal extraction (thermal variation + shoulder motion); band-pass filtering + autocorrelation/FFT for RR estimation	r=0.83 vs. belt; MAE ≈2.4; RMSE ≈2.4	No	Dual-ROI thermal approach combining nasal temperature and shoulder motion

## Data Availability

The datasets generated and analyzed during the current study are available from the corresponding author on reasonable request.

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
