# Peer review of "Adaptive Thermal Imaging Signal Analysis for Real-Time Non-Invasive Respiratory Rate Monitoring"

_sensors, 2026, doi:10.3390/s26010278_

Round 1

Reviewer 1 Report

Comments and Suggestions for Authors

Review Comment: Major Revision

This paper presents a highly integrated, non-contact respiratory rate monitoring embedded system based on thermal imaging technology. The combination of a lightweight YOLOv8n nostril detector, a Kalman filter-based tracking strategy, and an adaptive MAD hysteresis phase detection algorithm constitutes a robust technical contribution, particularly optimized for edge deployment. Experimental designs encompass relevant real-world scenarios, with accuracy falling within clinically significant ranges. However, several critical methodological shortcomings currently undermine the robustness and generalizability of the findings. In the author's opinion, the paper is not suitable for acceptance at this stage.

  1. For studies aimed at verifying system robustness, relying solely on 10 healthy subjects (with a narrow age range of 33.3 ± 4.38 years) is severely inadequate. This significantly limits the generalizability of the findings. It is recommended to expand the subject pool to include a more diverse sample.
  2. The paper lacks quantitative comparisons with baseline methods using the same dataset. The comparison in Table 5 is descriptive and based on data from different literature, which lacks persuasiveness. It is recommended to add at least 1-2 baseline methods for comparative experiments.
  3. This paper lacks an in-depth analysis of the system's failure modes under complex scenarios. For instance, it attributes the increased error under “soft-spoken” conditions solely to “facial movements” without examining the specific movement patterns causing signal degradation. Similarly, the analysis of performance decline with increasing distance remains at the level of “reduced pixels in the target area,” failing to quantify the corresponding decrease in signal-to-noise ratio.
  4. The introduction and related work sections should strengthen the argument regarding the necessity of embedded edge computing in respiratory rate monitoring or biomedical monitoring, citing recent relevant research findings.
  5. In the discussion section, it is recommended to directly compare key performance indicators with recently published respiratory monitoring studies to highlight the competitiveness of this research.
  6. Minor grammatical and structural errors exist, such as “which based on” should be “which is based on.” A comprehensive language edit and polish is recommended.

Author Response

Comments 1: For studies aimed at verifying system robustness, relying solely on 10 healthy subjects (with a narrow age range of 33.3 ± 4.38 years) is severely inadequate. This significantly limits the generalizability of the findings. It is recommended to expand the subject pool to include a more diverse sample.
Response:  We thank the reviewer for highlighting this important point. We agree that the use of ten healthy adults within a narrow age range limits the generalizability of the findings. As this work is designed as a feasibility study, the initial evaluation was intentionally conducted on a small, homogeneous cohort. The Discussion section has now been revised (page 27, lines 704–707) to explicitly acknowledge this limitation and to outline plans for expanded clinical validation involving elderly adults and patients with respiratory conditions.

Comments 2: The paper lacks quantitative comparisons with baseline methods using the same dataset. The comparison in Table 5 is descriptive and based on data from different literature, which lacks persuasiveness. It is recommended to add at least 1-2 baseline methods for comparative experiments.
Response: We appreciate this important suggestion. In accordance with this comment, we implemented two baseline methods, peak detection and FFT spectral estimation, directly on the same real-time experimental recordings. Their MAE and RMSE across all six experimental conditions were computed for all subjects.

A new Table 4 has been added (page 23) reporting:

  • Proposed method vs Peak vs FFT
  • Mean ± SD MAE and RMSE
  • Per-condition comparisons

Additionally, explanatory text referencing these results has been included in the manuscript (page 23, lines 555–557 and page 25, lines 607–615).

Comments 3: This paper lacks an in-depth analysis of the system's failure modes under complex scenarios. For instance, it attributes the increased error under “soft-spoken” conditions solely to “facial movements” without examining the specific movement patterns causing signal degradation. Similarly, the analysis of performance decline with increasing distance remains at the level of “reduced pixels in the target area,” failing to quantify the corresponding decrease in signal-to-noise ratio.
Response: We thank the reviewer for this constructive point. A detailed failure-mode analysis has now been added in the Discussion (page 26, lines 645–656). Specifically:

  • Soft-speech degradation is attributed not only to general motion but upper-lip deformation, transient nostril occlusion, and thermal discontinuities.
  • Distance-related degradation is quantitatively linked to a 597→165 px² ROI shrinkage, reduced thermal gradient resolution, lower oscillation amplitude, and increased quantization noise sensitivity.

This provides the deeper quantitative explanation requested.

Comments 4: The introduction and related work sections should strengthen the argument regarding the necessity of embedded edge computing in respiratory rate monitoring or biomedical monitoring, citing recent relevant research findings.
Response: We agree. Additional supporting literature on edge computing in biomedical sensing has been incorporated into the Introduction (page 2, lines 59-67) and Related work (page 4, lines 158–166).

Comments 5: In the discussion section, it is recommended to directly compare key performance indicators with recently published respiratory monitoring studies to highlight the competitiveness of this research.
Response: A new paragraph comparing Table 6 studies has been added in Discussion (page 26–27, lines 668-680). The revised text explicitly states: How our accuracy (MAE 0.57 ± 0.36 BPM) compares with Mozafari et al. (~1.6 BPM), Nakai et al. (~2.4 BPM), Gioia et al. (R² ≈ 0.61) and classical spectral methods (0.66–1.8 BPM). This comparison showcases competitiveness and feasibility of our approach.

Comments 6: Minor grammatical and structural errors exist, such as “which based on” should be “which is based on.” A comprehensive language edit and polish is recommended.
Response: A full language revision has been conducted to correct grammatical and structural issues throughout the manuscript.

Reviewer 2 Report

Comments and Suggestions for Authors

General comment:

This manuscript introduces a non-invasive respiratory rate monitoring system based on thermal imaging and advanced signal processing techniques. The proposal was tested under real conditions, achieving promising results, as evidenced by two well-known performance metrics. The main contribution is the use of a low-resolution thermal sensor coupled with estimation algorithms running on an edge device. This is an outstanding paper that deserves publication. I appreciate reading this kind of high-quality paper.

A couple of comments should be addressed to strengthen the paper’s readability.

Comment 1:

Could you give a brief rationale behind the state-space model defined in (15) and (16)? Have you proven observability and reachability conditions for your model? What about stability? Is the matrix F Hurwitz?

Comment 2:

The authors claim that the computational burden is light as it can run on an edge device. However, quantifying the computational complexity is missing from the manuscript. Could you elaborate on that?

Author Response

Comments 1: Could you give a brief rationale behind the state-space model defined in (15) and (16)? Have you proven observability and reachability conditions for your model? What about stability? Is the matrix F Hurwitz?

Response: Thank you for pointing this out. A detailed explanation has been added to page 9, lines 289–299. We clarified:

  • The model uses a constant-velocity formulation standard in visual tracking,
  • Observability is ensured because (F, H) has full rank for Δt > 0,
  • Controllability is not required since the filter performs estimation,

F is not Hurwitz because discrete-time constant-velocity systems are marginally stable, which is appropriate for this application.

Comments 2: The authors claim that the computational burden is light as it can run on an edge device. However, quantifying the computational complexity is missing from the manuscript. Could you elaborate on that?

Response: This comment has been fully addressed by adding a detailed 60-second runtime profiling experiment on the Jetson Orin Nano. The results are included in Section 4.1 and Table 1 (page 16, lines 434–448).

We now report:

  • Per-frame processing time (mean 65.2 ms, p95 85.3 ms)
  • Module-level runtime breakdown (YOLO 54.5%, tracking, filtering, ROI extraction)
  • System resource usage (CPU 42.5%, GPU 68.2%)
  • Throughput statistics (22.5 FPS real-time)

Reviewer 3 Report

Comments and Suggestions for Authors

The manuscript is generally very well prepared. The main content has enough main text, figures, and equations to support the results. However, I recommend increasing the font size of the axis titles and legends to improve readability.

For example, the axis titles and legends in Figures 5, 6, 7, 11, and 15 are quite small. For Figure 15, you do not need to put the legend inside the plot region and repeatedly show it. The font size in Figure 7 is too small. Although they become clear when zoomed in on a screen, they may still be difficult to read in a printed version of the manuscript.

Author Response

Comments 1:  The manuscript is generally very well prepared. The main content has enough main text, figures, and equations to support the results. However, I recommend increasing the font size of the axis titles and legends to improve readability.

For example, the axis titles and legends in Figures 5, 6, 7, 11, and 15 are quite small. For Figure 15, you do not need to put the legend inside the plot region and repeatedly show it. The font size in Figure 7 is too small. Although they become clear when zoomed in on a screen, they may still be difficult to read in a printed version of the manuscript.

Response: We thank the reviewer for this helpful suggestion. The axis titles, axis labels, and legends in all relevant figures have been enlarged and adjusted to improve readability. 

Reviewer 4 Report

Comments and Suggestions for Authors

This manuscript presents an innovative approach that uses thermal images and a YOLO-based model to identify breathing phases and calculate patients' respiratory rates across different experimental conditions. The concept is creative and addresses a critical clinical or assisted living need.

While the authors reported a good, robust model performance for the proposed approach, I recommend a major revision that can improve the paper quality:

  1. It would be helpful if the authors added research questions to their Introduction section.
  2. The experimental study was conducted on ten healthy participants. For this reason, can the authors add some more comments on how the results were validated? How was the model’s overfitting excluded?
  3. The study has limited clinical validation, and it was not performed across diverse populations, including elderly patients and individuals with respiratory pathologies, such as COPD, asthma, or sleep apnoea. I recommend adding this point to the Discussion section.
  4. The research incorporated multiple thermal cameras to capture biometric patterns and spatial information, and there is no discussion of patient privacy. Did the study include a privacy impact assessment? In these terms, can the author compare their approach with genuinely anonymous sensing modalities such as radar or acoustic methods?

Author Response

Comments 1: It would be helpful if the authors added research questions to their Introduction section.

Response: We appreciate this suggestion. Three research questions (RQ1–RQ3) have now been added to the end of the Introduction (page 3, lines 108–119). They clarify scientific motivation and guide the structure of the paper.

Comments 2: The experimental study was conducted on ten healthy participants. For this reason, can the authors add some more comments on how the results were validated? How was the model’s overfitting excluded?

Response:  A dedicated explanation is added in the Discussion (page 27, lines 681–688):

  • No part of the RR estimation pipeline involves training; parameters are fixed.
  • The YOLO detector was trained on a separate 7,958-image dataset, with no overlap with subjects used for RR evaluation.
  • Ground truth was obtained via dual-rater manual annotation.

This confirms that overfitting to evaluation data is not possible.

Comments 3: The study has limited clinical validation, and it was not performed across diverse populations, including elderly patients and individuals with respiratory pathologies, such as COPD, asthma, or sleep apnoea. I recommend adding this point to the Discussion section.

Response: 

This limitation has been incorporated into the Discussion (page 27, lines 707–711), clearly stating:

“Consequently, elderly adults or individuals with respiratory conditions such as COPD, asthma, or sleep apnoea were not included, resulting in limited clinical validation and reduced population diversity. Future work will conduct broader clinical validation involving these patient groups, as well as individuals at risk of long-lie incidents, to ensure that the system performs reliably across diverse real-world populations.”

Comments 4: The research incorporated multiple thermal cameras to capture biometric patterns and spatial information, and there is no discussion of patient privacy. Did the study include a privacy impact assessment? In these terms, can the author compare their approach with genuinely anonymous sensing modalities such as radar or acoustic methods?

Response:  An expanded privacy discussion has been added in Discussion (page 25, lines 616–628). It now explains:

  • Thermal imaging contains no facial texture,
  • Our ROI is restricted to nostril patch, enhancing privacy,

Comparison with radar/acoustic: radar is more anonymous but lacks anatomical anchoring; thermal balances privacy with spatial specificity.

Round 2

Reviewer 4 Report

Comments and Suggestions for Authors

All comments were addressed, thank you.